# Image Quality Assessment Techniques Improve Training and Evaluation of Energy-Based Generative Adversarial Networks

## Abstract

We propose a new, multi-component energy function for energy-based Generative Adversarial Networks (GANs) based on methods from the image quality assessment literature. Our approach expands on the Boundary Equilibrium Generative Adversarial Network (BEGAN) by outlining some of the short-comings of the original energy and loss functions. We address these short-comings by incorporating an $l_1$ score, the Gradient Magnitude Similarity score, and a chrominance score into the new energy function. We then provide a set of systematic experiments that explore its hyper-parameters. We show that each of the energy function's components is able to represent a slightly different set of features, which require their own evaluation criteria to assess whether they have been adequately learned. We show that models using the new energy function are able to produce better image representations than the BEGAN model in predicted ways.

## 1 Introduction

### 1.1 Improving learned representations for generative modeling

Radford et al. (2015) demonstrated that Generative Adversarial Networks (GANs) are a good unsupervised technique for learning representations of images for the generative modeling of 2D images. Since then, a number of improvements have been made. First, Zhao et al. (2016) modified the error signal of the deep neural network from the original, single parameter criterion to a multi-parameter criterion using auto-encoder reconstruction loss. Berthelot et al. (2017) then further modified the loss function from a hinge loss to the Wasserstein distance between loss distributions. For each modification, the proposed changes improved the resulting output to visual inspection (see Appendix A Figure 4, Row 1 for the output of the most recent, BEGAN model). We propose a new loss function, building on the changes of the BEGAN model (called the scaled BEGAN GMSM) that further modifies the loss function to handle a broader range of image features within its internal representation.

### 1.2 Generative Adversarial Networks

Generative Adversarial Networks are a form of two-sample or hypothesis testing that uses a classifier, called a *discriminator*, to distinguish between observed (training) data and data generated by the model or *generator*. Training is then simplified to a competing (i.e., adversarial) objective between the discriminator and generator, where the discriminator is trained to better differentiate training from generated data, and the generator is trained to better trick the discriminator into thinking its generated data is real. The convergence of a GAN is achieved when the generator and discriminator reach a Nash equilibrium, from a game theory point of view (Zhao et al., 2016).

In the original GAN specification, the task is to learn the generator's distribution $p_G$ over data $\boldsymbol{x}$ (Goodfellow et al., 2014). To accomplish this, one defines a generator function $G(\boldsymbol{z}; \theta_G)$, which produces an image using a noise vector $\boldsymbol{z}$ as input, and $G$ is a differentiable function with parameters $\theta_G$. The discriminator is then specified as a second function $D(\boldsymbol{x}; \theta_D)$ that outputs a scalar representing the probability that $\boldsymbol{x}$ came from the data rather than $p_G$. $D$ is then trained to maximize the probability of assigning the correct labels to the data and the image output of $G$ while $G$

is trained to minimize the probability that $D$ assigns its output to the fake class, or $1 - D(G(z))$. Although $G$ and $D$ can be any differentiable functions, we will only consider deep convolutional neural networks in what follows.

Zhao et al. (2016) initially proposed a shift from the original single-dimensional criterion—the scalar class probability—to a multidimensional criterion by constructing $D$ as an autoencoder. The image output by the autoencoder can then be directly compared to the output of $G$ using one of the many standard distance functions (e.g., $l_1$ norm, mean square error). However, Zhao et al. (2016) also proposed a new interpretation of the underlying GAN architecture in terms of an energy-based model (LeCun et al., 2006).

### 1.3 ENERGY-BASED GENERATIVE ADVERSARIAL NETWORKS

The basic idea of energy-based models (EBMs) is to map an input space to a single scalar or set of scalars (called its "energy") via the construction of a function (LeCun et al., 2006). Learning in this framework modifies the energy surface such that desirable pairings get low energies while undesirable pairings get high energies. This framework allows for the interpretation of the discriminator ($D$) as an energy function that lacks any explicit probabilistic interpretation (Zhao et al., 2016). In this view, the discriminator is a trainable cost function for the generator that assigns low energy values to regions of high data density and high energy to the opposite. The generator is then interpreted as a trainable parameterized function that produces samples in regions assigned low energy by the discriminator. To accomplish this setup, Zhao et al. (2016) first define the discriminator's energy function as the mean square error of the reconstruction loss of the autoencoder, or:

$$\mathcal{E}_D(x) = ||Decoder(Encoder(x)) - x|| \tag{1}$$

Zhao et al. (2016) then define the loss function for their discriminator using a form of margin loss.

$$\mathcal{L}_D(x, z) = \mathcal{E}_D(x) + [m - \mathcal{E}_D(G(z))]^+ \tag{2}$$

where $m$ is a constant and $[\cdot]^+ = max(0, \cdot)$. They define the loss function for their generator:

$$\mathcal{L}_G(z) = \mathcal{E}_D(G(z)) \tag{3}$$

The authors then prove that, if the system reaches a Nash equilibrium, then the generator will produce samples that cannot be distinguished from the dataset. Problematically, simple visual inspection can easily distinguish the generated images from the dataset.

### 1.4 DEFINING THE PROBLEM

It is clear that, despite the mathematical proof of Zhao et al. (2016), humans can distinguish the images generated by energy-based models from real images. There are two direct approaches that could provide insight into this problem, both of which are outlined in the original paper. The first approach that is discussed by Zhao et al. (2016) changes Equation 2 to allow for better approximations than $m$. The BEGAN model takes this approach. The second approach addresses Equation 1, but was only implicitly addressed when (Zhao et al., 2016) chose to change the original GAN to use the reconstruction error of an autoencoder instead of a binary logistic energy function. We chose to take the latter approach while building on the work of BEGAN.

Our main contributions are as follows:

- An energy-based formulation of BEGAN's solution to the visual problem.
- An energy-based formulation of the problems with Equation 1.
- Experiments that explore the different hyper-parameters of the new energy function.
- Evaluations that provide greater detail into the learned representations of the model.
- A demonstration that scaled BEGAN+GMSM can be used to generate better quality images from the CelebA dataset at 128x128 pixel resolution than the original BEGAN model in quantifiable ways.

## 2 IMPROVING THE ENERGY-BASED MODEL OF GANS

### 2.1 BOUNDARY EQUILIBRIUM GENERATIVE ADVERSARIAL NETWORKS

The Boundary Equilibrium Generative Adversarial Network (BEGAN) makes a number of modifications to the original energy-based approach. However, the most important contribution can be summarized in its changes to Equation 2. In place of the hinge loss, Berthelot et al. (2017) use the Wasserstein distance between the autoencoder reconstruction loss distributions of $G$ and $D$. They also add three new hyper-parameters in place of $m$: $k_t$, $\lambda_k$, and $\gamma$. Using an energy-based approach, we get the following new equation:

$$\mathcal{L}_D(x, z) = \mathcal{E}_D(x) - k_t \cdot \mathcal{E}_D(G(z)) \tag{4}$$

The value of $k_t$ is then defined as:

$$k_{t+1} = k_t + \lambda_k(\gamma \mathcal{E}_D(x) - \mathcal{E}_D(G(z))) \text{ for each } t \tag{5}$$

where $k_t \in [0, 1]$ is the emphasis put on $\mathcal{E}(G(z))$ at training step $t$ for the gradient of $\mathcal{E}_D$, $\lambda_k$ is the learning rate for $k$, and $\gamma \in [0, 1]$.

Both Equations 2 and 4 are describing the same phenomenon: the discriminator is doing well if either 1) it is properly reconstructing the real images or 2) it is detecting errors in the reconstruction of the generated images. Equation 4 just changes how the model achieves that goal. In the original equation (Equation 2), we punish the discriminator ($\mathcal{L}_D \to \infty$) when the generated input is doing well ($\mathcal{E}_D(G(z)) \to 0$). In Equation 4, we reward the discriminator ($\mathcal{L}_D \to 0$) when the generated input is doing poorly ($\mathcal{E}_D(G(z)) \to \infty$).

What is also different between Equations 2 and 4 is the way their boundaries function. In Equation 2, $m$ only acts as a one directional boundary that removes the impact of the generated input on the discriminator if $\mathcal{E}_D(G(z)) > m$. In Equation 5, $\gamma \mathcal{E}_D(x)$ functions in a similar but more complex way by adding a dependency to $\mathcal{E}_D(x)$. Instead of 2 conditions on either side of the boundary $m$, there are now four:

1. If $\gamma \mathcal{E}_D(x) > \mathcal{E}_D(G(z))$ and $\mathcal{E}_D(G(z)) \to \infty$, then $\mathcal{L}_D \to 0$ and it is accelerating as $k_t \to 1$.
2. If $\gamma \mathcal{E}_D(x) > \mathcal{E}_D(G(z))$ and $\mathcal{E}_D(G(z)) \to 0$, then $\mathcal{L}_D \to \mathcal{E}_D(x)$ and it is accelerating as $k_t \to 1$.
3. If $\gamma \mathcal{E}_D(x) < \mathcal{E}_D(G(z))$ and $\mathcal{E}_D(G(z)) \to \infty$, then $\mathcal{L}_D \to 0$ and it is decelerating as $k_t \to 0$.
4. If $\gamma \mathcal{E}_D(x) < \mathcal{E}_D(G(z))$ and $\mathcal{E}_D(G(z)) \to 0$, then $\mathcal{L}_D \to \infty$ and it is decelerating as $k_t \to 0$.

The optimal condition is condition 1 Berthelot et al. (2017). Thus, the BEGAN model tries to keep the energy of the generated output approaching the limit of the energy of the real images. As the latter will change over the course of learning, the resulting boundary dynamically establishes an equilibrium between the energy state of the real and generated input.[1]

It is not particularly surprising that these modifications to Equation 2 show improvements. Zhao et al. (2016) devote an appendix section to the correct selection of $m$ and explicitly mention that the "balance between... *real* and *fake* samples[s]" (italics theirs) is crucial to the correct selection of $m$. Unsurprisingly, a dynamically updated parameter that accounts for this balance is likely to be the best instantiation of the authors' intuitions and visual inspection of the resulting output supports this (see Berthelot et al., 2017). We chose a slightly different approach to improving the proposed loss function by changing the original energy function (Equation 1).

### 2.2 FINDING A NEW ENERGY FUNCTION VIA IMAGE QUALITY ASSESSMENT

In the original description of the energy-based approach to GANs, the energy function was defined as the mean square error (MSE) of the reconstruction loss of the autoencoder (Equation 1). Our first

---

[1]For a much more detailed and formal account that is beyond the scope of the current paper, see (Berthelot et al., 2017).

insight was a trivial generalization of Equation 1:

$$\mathcal{E}(x) = \delta(D(x), x) \tag{6}$$

where $\delta$ is some distance function. This more general equation suggests that there are many possible distance functions that could be used to describe the reconstruction error and that the selection of $\delta$ is itself a design decision for the resulting energy and loss functions. Not surprisingly, an entire field of study exists that focuses on the construction of similar $\delta$ functions in the image domain: the field of image quality assessment (IQA).

The field of IQA focuses on evaluating the quality of digital images (Wang & Bovik, 2006). IQA is a rich and diverse field that merits substantial further study. However, for the sake of this paper, we want to emphasize three important findings from this field. First, distance functions like $\delta$ are called full-reference IQA (or FR-IQA) functions because the reconstruction ($D(x)$) has a 'true' or undistorted reference image ($x$) which it can be evaluated from Wang et al. (2004). Second, IQA researchers have known for a long time that MSE is a poor indicator of image quality (Wang & Bovik, 2006). And third, there are numerous other functions that are better able to indicate image quality. We explain each of these points below.

One way to view the FR-IQA approach is in terms of a reference and distortion vector. In this view, an image is represented as a vector whose dimensions correspond with the pixels of the image. The reference image sets up the initial vector from the origin, which defines the original, perfect image. The distorted image is then defined as another vector defined from the origin. The vector that maps the reference image to the distorted image is called the *distortion vector* and FR-IQA studies how to evaluate different types of distortion vectors. In terms of our energy-based approach and Equation 6, the distortion vector is measured by $\delta$ and it defines the surface of the energy function.

MSE is one of the ways to measure distortion vectors. It is based in a paradigm that views the loss of quality in an image in terms of the visibility of an error signal, which MSE quantifies. Problematically, it has been shown that MSE actually only defines the *length* of a distortion vector not its type (Wang & Bovik, 2006). For any given reference image vector, there are an entire hypersphere of other image vectors that can be reached by a distortion vector of a given size (i.e., that all have the same MSE from the reference image; see Figure 1).

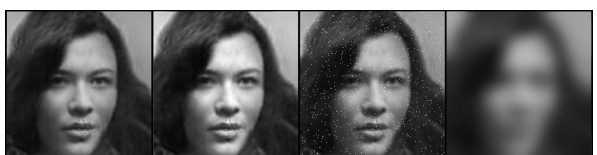

Figure 1: From left to right, the images are the original image, a contrast stretched image, an image with impulsive noise contamination, and a Gaussian smoothed image. Although these images differ greatly in quality, they all have the same MSE from the original image (about 400), suggesting that MSE is a limited technique for measuring image quality.

A number of different measurement techniques have been created that improve upon MSE (for a review, see Chandler, 2013). Often these techniques are defined in terms of the similarity ($S$) between the reference and distorted image, where $\delta = 1 - S$. One of the most notable improvements is the Structural Similarity Index (SSIM), which measures the similarity of the luminance, contrast, and structure of the reference and distorted image using the following similarity function:[2]

$$S(\boldsymbol{v}_d, \boldsymbol{v}_r) = \frac{2\boldsymbol{v}_d\boldsymbol{v}_r + C}{\boldsymbol{v}_d^2 + \boldsymbol{v}_r^2 + C} \tag{7}$$

where $v_d$ is the distorted image vector, $v_r$ is the reference image vector, C is a constant, and all multiplications occur element-wise Wang & Bovik (2006).[3] This function has a number of desirable

---

[2]The SSIM similarity function is reminiscent of the Dice-Sorensen distance function. It is worth noting that the Dice-Sorensen distance function does not satisfy the triangle inequality for sets Gragera & Suppakitpaisarn (2016). Since sets are a restricted case for Equation 7, where all the values are either 0 or 1, we can conclude that the corresponding distance of Equation 7 also fails to satisfy the triangle inequality. Consequently, it is not a true distance *metric*.

[3]We use $C = 0.0026$ following the work on cQS described below Gupta et al. (2017).

features. It is symmetric (i.e., $S(v_d, v_r) = S(v_r, v_d)$), bounded by 1 (and 0 for $x > 0$), and it has a unique maximum of 1 only when $v_d = v_r$. Although we chose not to use SSIM as our energy function ($\delta$) as it can only handle black-and-white images, its similarity function (Equation 7) informs our chosen technique.

The above discussion provides some insights into why visual inspection fails to show this correspondence between real and generated output of the resulting models, even though Zhao et al. (2016) proved that the generator should produce samples that cannot be distinguished from the dataset. The original proof by Zhao et al. (2016) did not account for Equation 1. Thus, when Zhao et al. (2016) show that their generated output should be indistinguishable from real images, what they are actually showing is that it should be indistinguishable from the real images *plus* some residual distortion vector described by $\delta$. Yet, we have just shown that MSE (the author's chosen $\delta$) can only constrain the length of the distortion vector, not its type. Consequently, it is entirely possible for two systems using MSE for $\delta$ to have both reached a Nash equilibrium, have the same energy distribution, and yet have radically different internal representations of the learned images. The energy function is as important as the loss function for defining the data distribution.

## 2.3 A NEW ENERGY FUNCTION

Rather than assume that any one distance function would suffice to represent all of the various features of real images, we chose to use a multi-component approach for defining $\delta$. In place of the luminance, contrast, and structural similarity of SSIM, we chose to evaluate the $l_1$ norm, the gradient magnitude similarity score (GMS), and a chrominance similarity score (Chrom). We outline the latter two in more detail below.

The GMS score and chrom scores derive from an FR-IQA model called the color Quality Score (cQS; Gupta et al., 2017). The cQS uses GMS and chrom as its two components. First, it converts images to the YIQ color space model. In this model, the three channels correspond to the luminance information (Y) and the chrominance information (I and Q). Second, GMS is used to evaluate the local gradients across the reference and distorted images on the luminance dimension in order to compare their edges. This is performed by convolving a $3 \times 3$ Sobel filter in both the horizontal and vertical directions of each image to get the corresponding gradients. The horizontal and vertical gradients are then collapsed to the gradient magnitude of each image using the Euclidean distance.[4] The similarity between the gradient magnitudes of the reference and distorted image are then compared using Equation 7. Third, Equation 7 is used to directly compute the similarity between the I and Q color dimensions of each image. The mean is then taken of the GMS score (resulting in the GMS*M* score) and the combined I and Q scores (resulting in the Chrom score).

In order to experimentally evaluate how each of the different components contribute to the underlying image representations, we defined the following, multi-component energy function:

$$\mathcal{E}_{\mathcal{D}} = \frac{\sum_{\delta \in \mathcal{D}} \delta(D(x), x) \beta_d}{\sum_{\delta \in \mathcal{D}} \beta_d} \tag{8}$$

where $\beta_d$ is the weight that determines the proportion of each $\delta$ to include for a given model, and $\mathcal{D}$ includes the $l_1$ norm, GMSM, and the chrominance part of cQS as individual $\delta$s. In what follows, we experimentally evaluate each of the energy function components($\beta$) and some of their combinations.

## 3 EXPERIMENTS

### 3.1 METHOD

We conducted extensive quantitative and qualitative evaluation on the CelebA dataset of face images Liu et al. (2015). This dataset has been used frequently in the past for evaluating GANs Radford et al. (2015); Zhao et al. (2016); Chen et al. (2016); Liu & Tuzel (2016). We evaluated 12 different models in a number of combinations (see Table 1). They are as follows. Models 1, 7, and 11 are the original BEGAN model. Models 2 and 3 only use the GMSM and chrominance distance functions, respectively. Models 4 and 8 are the BEGAN model plus GMSM. Models 5 and 9 use all three

---

[4]For a detailed outline of the original GMS function, see Xue et al. (2014).

| Model # | Model Parameters | | | | |
|---|---|---|---|---|---|
| | Size | $\gamma$ | $l_1$ | GMSM | Chrom |
| 01 | 64 | 0.5 | 1 | 0 | 0 |
| 02 | 64 | 0.5 | 0 | 1 | 0 |
| 03 | 64 | 0.5 | 0 | 0 | 1 |
| 04 | 64 | 0.5 | 1 | 1 | 0 |
| 05 | 64 | 0.5 | 1 | 1 | 1 |
| 06 | 64 | 0.5 | 2 | 1 | 0 |
| 07 | 64 | 0.7 | 1 | 0 | 0 |
| 08 | 64 | 0.7 | 1 | 1 | 0 |
| 09 | 64 | 0.7 | 1 | 1 | 1 |
| 10 | 64 | 0.7 | 2 | 1 | 0 |
| 11 | 128 | 0.7 | 1 | 0 | 0 |
| 12 | 128 | 0.7 | 2 | 1 | 0 |

Table 1: Models and their corresponding model distance function parameters. The $l_1$, GMSM, and Chrom parameters are their respective $\beta_d$ values from Equation 8.

distance functions (BEGAN+GMSM+Chrom). Models 6, 10, and 12 use a 'scaled' BEGAN model ($\beta_{l_1} = 2$) with GMSM. All models with different model numbers but the same $\beta_d$ values differ in their $\gamma$ values or the output image size.

## 3.2 SETUP

All of the models we evaluate in this paper are based on the architecture of the BEGAN model Berthelot et al. (2017).[5] We trained the models using Adam with a batch size of 16, $\beta_1$ of 0.9, $\beta_2$ of 0.999, and an initial learning rate of 0.00008, which decayed by a factor of 2 every 100,000 epochs.

Parameters $k_t$ and $k_0$ were set at 0.001 and 0, respectively (see Equation 5). The $\gamma$ parameter was set relative to the model (see Table 1).

Most of our experiments were performed on $64 \times 64$ pixel images with a single set of tests run on $128 \times 128$ images. The number of convolution layers were 3 and 4, respectively, with a constant down-sampled size of $8 \times 8$. We found that the original size of 64 for the input vector ($N_z$) and hidden state ($N_h$) resulted in modal collapse for the models using GMSM. However, we found that this was fixed by increasing the input size to 128 and 256 for the 64 and 128 pixel images, respectively. We used $N_z = 128$ for all models except 12 (scaled BEGAN+GMSM), which used 256. $N_z$ always equaled $N_h$ in all experiments.

Models 2-3 were run for 18,000 epochs, 1 and 4-10 were run for 100,000 epochs, and 11-12 were run for 300,000 epochs. Models 2-4 suffered from modal collapse immediately and 5 (BEGAN+GMSM+Chrom) collapsed around epoch 65,000 (see Appendix A Figure 4 rows 2-5).

## 3.3 EVALUATIONS

We performed two evaluations. First, to evaluate whether and to what extent the models were able to capture the relevant properties of each associated distance function, we compared the mean and standard deviation of the error scores. We calculated them for each distance function over all epochs of all models. We chose to use the mean rather than the minimum score as we were interested in how each model performs as a whole, rather than at some specific epoch. All calculations use the distance, or one minus the corresponding similarity score, for both the gradient magnitude and chrominance values.

Reduced pixelation is an artifact of the intensive scaling for image presentation (up to $4\times$). All images in the qualitative evaluations were upscaled from their original sizes using cubic image sampling so that they can be viewed at larger sizes. Consequently, the apparent smoothness of the scaled images is not a property of the model.

---

[5]The code for the model and all related experiments are currently available on Github. Links will be included post-review.

| Model # | Discriminator Loss Statistics | | | | | |
| | $l_1$ | | GMSM | | Chrom | |
| | $M$ | $\sigma$ | $M$ | $\sigma$ | $M$ | $\sigma$ |
|---|---|---|---|---|---|---|
| 01 | **0.12** | 0.02 | 0.24 | 0.02 | 0.68 | 0.08 |
| 02 | 0.90 | 0.09 | **0.17** | 0.03 | 0.99 | 0.01 |
| 03 | 0.51 | 0.02 | 0.52 | 0.04 | **0.46** | 0.08 |
| 04 | 0.11 | 0.01 | **0.16** | 0.02 | 0.75 | 0.07 |
| 05 | 0.13 | 0.02 | 0.20 | 0.02 | **0.41** | 0.05 |
| 06 | **0.10** | 0.01 | 0.17 | 0.02 | 0.69 | 0.07 |
| 07 | **0.10** | 0.01 | 0.22 | 0.01 | 0.63 | 0.08 |
| 08 | 0.11 | 0.01 | *0.16* | 0.02 | 0.83 | 0.07 |
| 09 | 0.13 | 0.02 | 0.20 | 0.02 | *0.42* | 0.06 |
| 10 | **0.10** | 0.01 | 0.17 | 0.02 | 0.72 | 0.08 |
| 11 | 0.09 | 0.01 | 0.29 | 0.01 | **0.58** | 0.08 |
| 12 | *0.08* | 0.02 | **0.17** | 0.02 | 0.63 | 0.08 |

Table 2: Lists the models, their discriminator mean error scores, and their standard deviations for the $l_1$, GMSM, and chrominance distance functions over all training epochs. Bold values show the best scores for similar models. Double lines separate sets of similar models. Values that are both bold and italic indicate the best scores overall, excluding models that suffered from modal collapse. These results suggest that model training should be customized to emphasize the relevant components.

## 3.4 RESULTS

GANs are used to generate different types of images. Which image components are important depends on the domain of these images. Our results suggest that models used in any particular GAN application should be customized to emphasize the relevant components—there is not a one-size-fits-all component choice. We discuss the results of our four evaluations below.

### 3.4.1 MEANS AND STANDARD DEVIATIONS OF ERROR SCORES

Results were as expected: the three different distance functions captured different features of the underlying image representations. We compared all of the models in terms of their means and standard deviations of the error score of the associated distance functions (see Table 2). In particular, each of models 1-3 only used one of the distance functions and had the lowest error for the associated function (e.g., model 2 was trained with GMSM and has the lowest GMSM error score). Models 4-6 expanded on the first three models by examining the distance functions in different combinations. Model 5 (BEGAN+GMSM+Chrom) had the lowest chrominance error score and Model 6 (scaled BEGAN+GMSM) had the lowest scores for $l_1$ and GMSM of any model using a $\gamma$ of 0.5.

For the models with $\gamma$ set at 0.7, models 7-9 showed similar results to the previous scores. Model 8 (BEGAN+GMSM) scored the lowest GMSM score overall and model 9 (BEGAN+GMSM+Chrom) scored the lowest chrominance score of the models that did not suffer from modal collapse. For the two models that were trained to generate $128 \times 128$ pixel images, model 12 (scaled BE-GAN+GMSM) had the lowest error scores for $l_1$ and GMSM, and model 11 (BEGAN) had the lowest score for chrominance. Model 12 had the lowest $l_1$ score, overall.

### 3.4.2 VISUAL COMPARISON OF SIMILARITY SCORES

Subjective visual comparison of the gradient magnitudes in column S of Figure 2 shows there are more black pixels for model 11 (row 11D) when comparing real images before and after autoencoding. This indicates a lower similarity in the autoencoder. Model 12 (row 12D) has a higher similarity between the original and autoencoded real images as indicated by fewer black pixels. This pattern continues for the generator output (rows 11G and 12G), but with greater similarity between the gradients of the original and autoencoded images than the real images (i.e., fewer black pixels overall).

The visual comparison of chrominance and related similarity score also weakly supported our hypotheses (see Figure 3). All of the models show a strong ability to capture the I dimension (blue-red)



Figure 2: Comparison of the gradient (edges in the image) for models 11 (BEGAN) and 12 (scaled BEGAN+GMSM), where O is the original image, A is the autoencoded image, OG is the gradient of the original image, AG is the gradient of the autoencoded image, and S is the gradient magnitude similarity score for the discriminator (D) and generator (G). White equals greater similarity (better performance) and black equals lower similarity for the final column.

of the YIQ color space, but only model 9 (BEGAN+GMSM+Chrom) is able to accurately capture the relevant information in the Q dimension (green-purple).

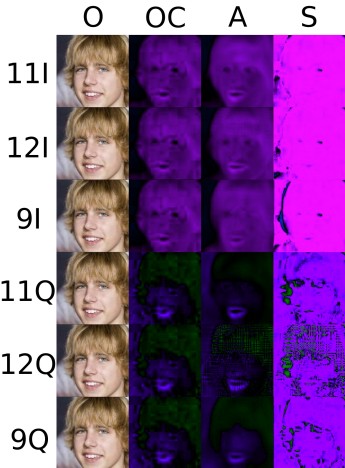

Figure 3: Comparison of the chrominance for models 9 (BEGAN+GMSM+Chrom), 11 (BEGAN) and 12 (scaled BEGAN+GMSM), where O is the original image, OC is the original image in the corresponding color space, A is the autoencoded image in the color space, and S is the chrominance similarity score. I and Q indicate the (blue-red) and (green-purple) color dimensions, respectively. All images were normalized relative to their maximum value to increase luminance. Note that pink and purple approximate a similarity of 1, and green and blue approximate a similarity of 0 for I and Q dimensions, respectively. The increased gradient 'speckling' of model 12Q suggests an inverse relationship between the GMSM and chrominance distance functions.

## 4  OUTLOOK

We bring an energy-based formulation to the BEGAN model and some of the problems of the energy function originally proposed in Zhao et al. (2016). We proposed a new, multi-component energy function on the basis of research from the Image Quality Assessment literature. The scaled BEGAN+GMSM model produces better image representations than its competitors in ways that can be measured using subjective evaluations of the associated features (e.g., luminance gradient similarity, chrominance similarity). For future work, we would like to extend this research to encompass other datasets and FR-IQA energy functions.

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

## A    VISUAL OUTPUT FROM ALL TWELVE MODELS

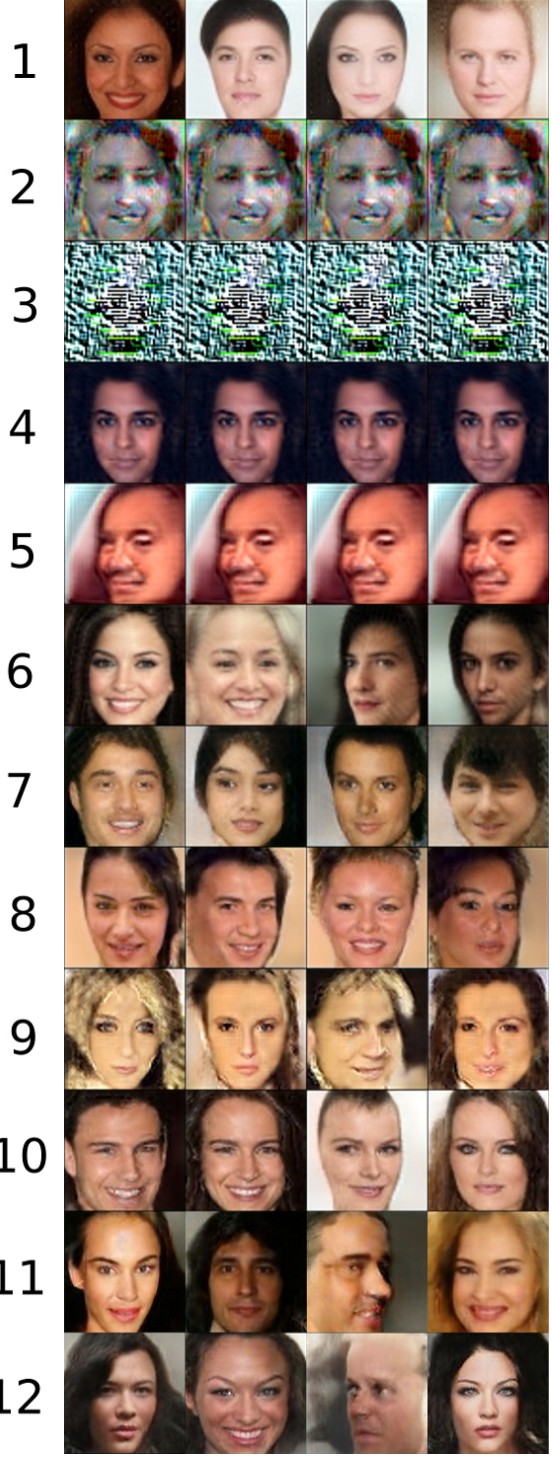

Figure 4: Four outputs of each of the generators of all 12 models. The best images for each model were hand-picked. The first row is model 1, which corresponds with the original BEGAN model. Rows 2-12 represent our experiments. Each cell represents the output of a random sample.

## B  FURTHER EVALUATIONS

### B.1  DIVERSITY OF LATENT SPACE

Further evidence that the models can generalize, and not merely memorize the input, can be seen in the linear interpolations in the latent space of $z$. In Figure 5 models 11 (BEGAN) and 12 (scaled BEGAN+GMSM) show smooth interpolation in gender, rotation, facial expression, hairstyle, and angle of the face.

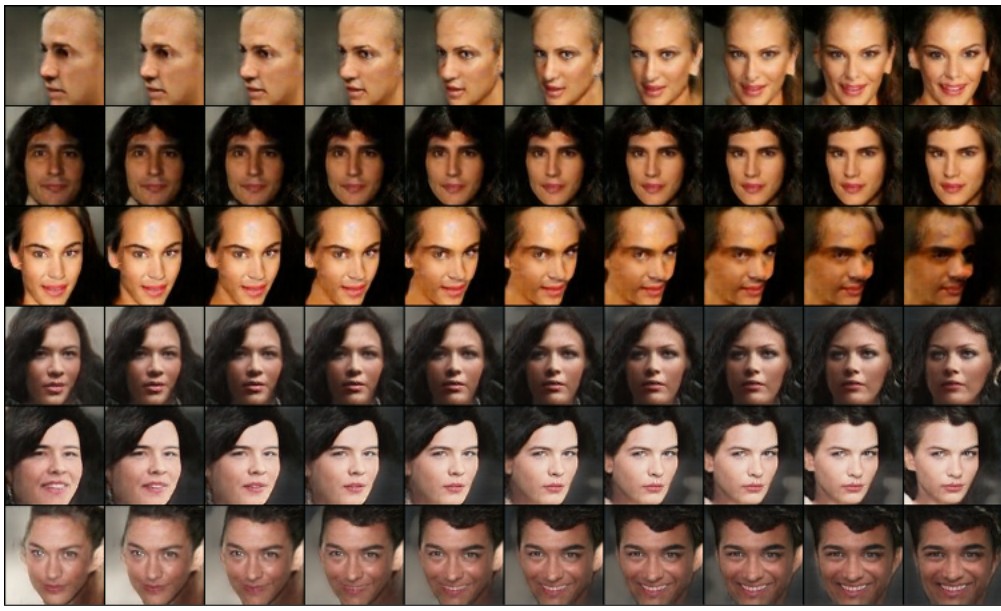

Figure 5: The new distance functions did not affect image diversity in the latent space of $z$. The top three rows are linear interpolations from model 11 (BEGAN) and the bottom three are from model 12 (scaled BEGAN+GMSM).

### B.2  THE BEGAN CONVERGENCE MEASURE

We compared the convergence measure scores for models 11 and 12 across all 300,000 epochs (see Figure 6; Berthelot et al. 2017). The convergence measure is defined as follows

$$\mathcal{M}_{global} = \mathcal{E}_D(x) + |\gamma \mathcal{E}_D(x) - \mathcal{E}_D(G(z))| \tag{9}$$

where the energy function is defined as per Equation 8. Due to the variance in this measure, we applied substantial Gaussian smoothing ($\sigma = 0.9$) to enhance the main trends. The output of a single generated image is also included for every 40,000 epochs, starting with epoch 20,000 and ending on epoch 300,000. Model 11 showed better (greater) convergence over the 300,000 epochs (as indicated by a lower convergence measure score). Both models continue to show that the convergence measure correlates with better images as the models converge.

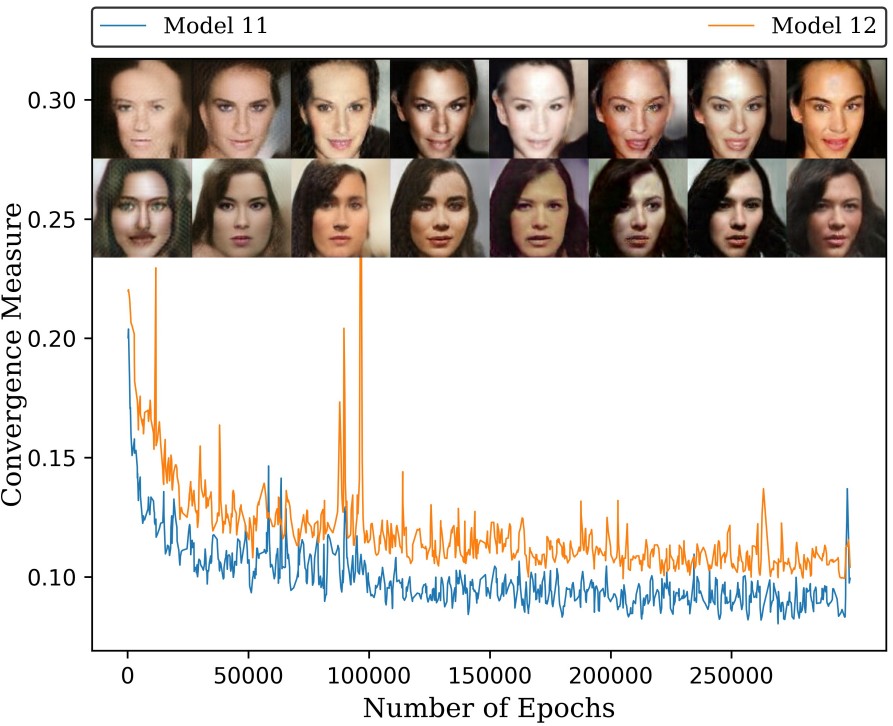

Figure 6: Measure of convergence and quality of the results for Models 11 (BEGAN; top images) and 12 (scaled BEGAN+GMSM; bottom images). The results were smoothed with a Gaussian with $\sigma = 0.9$. Images are displayed in 40,000 epoch increments starting with epoch 20,000 and going to 300,000. The output of earlier training epochs appear to be more youthful. As training proceeds, finer details are learned by the model, resulting in apparent increased age.

