# OpenReview forum: "Image Quality Assessment Techniques Improve Training and Evaluation of Energy-Based Generative Adversarial Networks"
_ICLR.cc/2018/Conference — Reject_

### Official Review · AnonReviewer1 · 2017-11-23
**A very technical paper with unclear significance.**

**Rating:** 5
**Confidence:** 3

**Review:**

Quick summary:
This paper proposes an energy based formulation to the BEGAN model and modifies it to include an image quality assessment based term. The model is then trained with CelebA under different parameters settings and results are analyzed.

Quality and significance:
This is quite a technical paper, written in a very compressed form and is a bit hard to follow. Mostly it is hard to estimate what is the contribution of the model and how the results differ from baseline models.

Clarity:
I would say this is one of the weak points of the paper - the paper is not well motivated and the results are not clearly presented.

Originality:
Seems original.

Pros:
* Interesting energy formulation and variation over BEGAN

Cons:
* Not a clear paper
* results are only partially motivated and analyzed

---

> ### Author Response · Authors · 2017-12-04
> **Clarifications**
>
> Thank you for your review and comments.
>
> Could you be more specific about what needs greater clarity?
>
> All of the models are modifications upon the original BEGAN model except model 1 (which is the original model). All of the modifications are based upon different hyper-parameter sets of equation 8 which are outlined in Table 1. Sections 2.2 and 2.3 motivate the modifications we made.

---

### Official Review · AnonReviewer2 · 2017-11-27
**Novelty of the paper is a bit restricted, and design choices appear to be lacking strong justifications.**

**Rating:** 5
**Confidence:** 3

**Review:**

This paper proposed some new energy function in the BEGAN (boundary equilibrium GAN framework), including l_1 score, Gradient magnitude similarity score, and chrominance score, which are motivated and borrowed from the image quality assessment techniques. These energy component in the objective function allows learning of different set of features and determination on whether the features are adequately represented. experiments on the using different hyper-parameters of the energy function, as well as visual inspections on the quality of the learned images, are presented.

It appears to me that the novelty of the paper is limited, in that the main approach is built on the existing BEGAN framework with certain modifications. For example, the new energy function in equation (4) larges achieves similar goal as the original energy (1) proposed by Zhao et. al (2016), except that the margin loss in (1) is changed to a re-weighted linear loss, where the dynamic weighting scheme of k_t is borrowed  from the work of Berthelot et. al (2017). It is not very clear why making such changes in the energy would supposedly make the results better, and no further discussions are provided.  On the other hand, the several energy component introduced are simply choices of the similarity measures as motivated from the image quality assessment, and there are probably a lot more in the literature whose application can not be deemed as a significant contribution to either theories or algorithm designs in GAN.

Many results from the experimental section rely on visual evaluations, such as in Figure~4 or 5; from these figures, it is difficult to clearly pick out the winning images. In Figure~5, for a fair  evaluation on the performance of model interploations, the same human model should be used for competing methods, instead of applying different human models and different interpolation tasks in different methods.

---

> ### Author Response · Authors · 2017-12-04
> **Clarifications**
>
> Thank you for your review and comments.
>
> Could you unpack what you mean by, "It is not very clear why making such changes in the energy would supposedly make the results better, and no further discussions are provided"? We explicitly state in section 2.1 that:
>
> "It is not particularly surprising that these modifications to Equation 2 show improvements. Zhao et al. (2016) devote an appendix section to the correct selection of m and explicitly mention that the “balance between... real and fake samples[s]” (italics theirs) is crucial to the correct selection of m. Unsurprisingly, a dynamically updated parameter that accounts for this balance is likely to be the best instantiation of the authors’ intuitions and visual inspection of the resulting output supports this (see Berthelot et al., 2017)."
>
> What kind of discussion would you have liked to see? If you're looking for a formal analysis, we would suggest reviewing Berthelot et al., (2017) and Arjovsky, Chintala, and Bottou (2017) for their discussions of the advantages of the Wasserstein distance over the alternatives. Section 5 bullet 3 in the latter explicitly addresses the differences between the original EBGAN margin loss and the Wasserstein distance, if you are interested. Sections 3.3 and 3.4 of the former address the equilibrium hyper-parameters of the BEGAN model (e.g., gamma, k_t).
>
> Could you also unpack what you mean by, "there are probably a lot more [similarity measures] in the [IQA] literature whose application can not be deemed as a significant contribution to either theories or algorithm designs in GAN"? We assume that you are not saying that the mere existence of other methods is damning to the scientific study of some subset of those methods. Please clarify how our modification of the energy-based formulation of GANs to emphasize a more important role for the energy function (generally assumed to be an l1 or l2 norm across many studies) is not a significant contribution to GAN research?
>
> Could you also unpack what you mean by, "human model"? We would like to clarify that the function of Figure 5 is to illustrate how image diversity has not been lost when using the new evaluation. It is not trying to show how one set of images are better than another.

---

### Official Review · AnonReviewer3 · 2017-12-01
**An incremental paper with moderately interesting results on a single dataset**

**Rating:** 6
**Confidence:** 3

**Review:**

Summary:
The paper extends the the recently proposed Boundary Equilibrium Generative Adversarial Networks (BEGANs), with the hope of generating images which are more realistic. In particular, the authors propose to change the energy function associated with the auto-encoder, from an L2 norm (a single number) to an energy function with multiple components. Their energy function is inspired by the structured similarity index (SSIM), and the three components they use are the L1 score, the gradient magnitude similarity score, and the chromium score. Using this energy function, the authors hypothesize, that it will force the generator to generate realistic images. They test their hypothesis on a single dataset, namely, the CelebA dataset.

Review:
While the idea proposed in the paper is somewhat novel and there is nothing obviously wrong about the proposed approach, I thought the paper is somewhat incremental. As a result I kind of question the impact of this result. My suspicion is reinforced by the fact that the experimental section is extremely weak. In particular the authors test their model on a single relatively straightforward dataset. Any reason why the authors did not try on other datasets involving natural images? As a result I feel that the title and the claims in the paper are somewhat misleading and premature: that the proposed techniques improves the training and evaluation of energy based gans.

Over all the paper is clearly written and easy to understand.

Based on its incremental nature and weak experiments, I'm on the margin with regards to its acceptance. Happy to change my opinion if other reviewers strongly think otherwise with good reason and are convinced about its impact.

---

> ### Author Response · Authors · 2017-12-04
> **Challenges of the BEGAN model**
>
> Thank you for your review and comments.
>
> We have been working on extending our research to include other datasets. The primary challenge is that the stock BEGAN model does rather poorly on datasets that do not have a lot of regular structure like the CelebA dataset. Consequently, we have preliminary results that are suggestive for MNIST and the msceleb dataset, but we've been unable to show any interesting results on Imagenet or the LSUN bedrooms dataset.
>
> Our suspicion is that these are issues with the stock network design/structure. We are currently working with an EBM-based modification of the model from "Progressive Growing of GANs for Improved Quality, Stability, and Variation" (this conference), which seems to replicate our results on other datasets. Our research is still very preliminary, though.
>
> We are curious what the 'correct' number of datasets is for a conference proceedings paper (which is extremely short). The original BEGAN paper only uses one, custom dataset. The EBGAN paper has 4 (if you count MNIST). The WGAN-GP paper has 2 plus some artificial datasets. Consequently, there doesn't seem to be any consensus in the community on this point.

---

### Decision · Program_Chairs · 2018-01-29
**ICLR 2018 Conference Acceptance Decision**

**Decision:**

Reject

**Comment:**

The paper received borderline-negative scores (6,5,5) with R1 and R2 having significant difficulty with the clarity of the paper. Although R3 was marginally positive, they pointed out that the experiments are "extremely weak". The AC look at the paper and agrees with R3 on this point. Therefore the paper cannot be accepted in its current form. The experiments and clarity need work before resubmission to another venue.